# Evaluation of the Effect of '*Candidatus* Liberibacter Solanacearum' Haplotypes in Tobacco Infection

**Julien G. Levy** [1],*[ID], **Azucena Mendoza-Herrera** [2], **Naveed Merchant** [3], **Katherine M. Berg-Falloure** [4], **Michael V. Kolomiets** [4][ID] and **Cecilia Tamborindeguy** [2],*

1. Department of Horticultural Sciences, Texas A&M University, AgriLife, College Station, TX 77843, USA
2. Department of Entomology, Texas A&M University, College Station, TX 77843, USA
3. Department of Statistics, Texas A&M University, College Station, TX 77843, USA
4. Department of Plant Pathology and Microbiology, Texas A&M University, College Station, TX 77843, USA
* Correspondence: julien.levy@ag.tamu.edu (J.G.L.); ctamborindeguy@ag.tamu.edu (C.T.)

**Abstract:** '*Candidatus* Liberibacter solanacearum' (Lso) is a phloem-limited bacterial plant pathogen infecting solanaceous plants in the Americas and New Zealand and is associated with diseases of apiaceous crops in Europe, Northern Africa, and the Middle East. This pathogen is also related to other Liberibacter species that infect other crops. In the USA, two haplotypes of Lso, LsoA and LsoB, are predominant and responsible for diseases in potato and tomato. Tobacco, *Nicotiana tabacum*, a model species to study plant defenses, is a host for Lso; therefore, the interaction between Lso and this host plant could be used to study Liberibacter−plant interactions. In this study, we characterized the infection associated with LsoA and LsoB in tobacco. Under laboratory conditions, LsoB caused more severe symptoms than LsoA, and LsoA and LsoB titers were dynamic during the 7 weeks of the experiment. We also measured SA and other metabolites, including oxylipins, at an early point of infection and found that SA was accumulated in plants infected with LsoB but not with LsoA; whereas ABA levels were reduced in LsoA- but not in LsoB-infected plants.

**Keywords:** psyllid; *Bactericera cockerelli* Sulč; *Candidatus* Liberibacter solanacearum; Lso haplotype; tomato; potato; zebra chip; solanaceous; salicylic acid

## 1. Introduction

'*Candidatus* Liberibacter solanacearum' (Lso) is a bacterial pathogen of solanaceous crops in the USA and the causal agent of zebra chip, a potato disease that costs several millions of dollars to the potato industry annually [1,2]. This pathogen is related to '*Candidatus* Liberibacter asiaticus' (CLas), the causal agent of huanglongbing (HLB), which is considered the most devastating disease affecting citrus worldwide [3]. CLas can infect citrus species, which ultimately results in tree death [4,5]. '*Candidatus* Liberibacter' pathogens are fastidious and are transmitted by psyllids. CLas and Lso have reduced genomes similar to other obligatory symbionts; these two pathogens separated from their common ancestor around 100 million years ago [6]. Management of the diseases caused by the Liberibacter pathogens relies mainly on the application of pesticides to reduce the vector populations, but this strategy has high costs for producers [2].

Currently, several Lso haplotypes have been identified. In the USA, there are two main Lso haplotypes associated with diseases in Solanaceae: LsoA and LsoB [7–10]. LsoF is a third haplotype potentially associated with zebra chip; it was recently identified from a single tuber originating from southern Oregon [11]. LsoA and LsoB are transmitted by the potato psyllid, *Bactericera cockerelli* Sulč, also known as the tomato psyllid. Other Lso haplotypes exist in different parts of the world associated with different plant and psyllid species [12–16]. For example, LsoC, LsoD, and LsoE are associated with apiaceous crops such as carrots in Europe, Northern Africa, and Israel, where they are transmitted by different carrot psyllid species such as *B. trigonica* and *B. nigricornis*.

The diseases caused by Lso in tomato and potato have been characterized. In potato, the diseases associated with LsoA and LsoB are relatively similar; both result in plant stunting, leaf chlorosis, decreased plant vigor, and early plant death, as well as the development of striping in the potato tubers when they are fried [17–19]. However, in tomato, the differences between LsoB and LsoA infection are more pronounced; while LsoA causes stunting and chlorosis, LsoB is significantly more damaging, additionally leading to plant death [20,21]. The disease caused by Lso in tobacco was discovered in 2013 in Nicaragua and Honduras [22,23]. In both countries, infected tobacco plants showed aerial symptoms similar to those described in potato and included apical leaf curling and stunting, overall chlorosis, young plant deformation with cabbage-like leaves, wilting, and internal vascular necrosis of stems and leaf petioles [22,23]. However, whether differences in symptoms associated with LsoA and LsoB infection in tobacco plants existed was not investigated. Tobacco is a crop, but it is also a model species to study plant−insect and plant−bacterial interactions. There is a wealth of knowledge and resources available for this plant. Therefore, differences in Lso pathogenicity associated with tobacco could have relevance not only from an epidemiological perspective but could also help elucidate the molecular basis of Lso pathogenicity and of the plant responses induced upon Lso infection. Therefore, in the present study, we characterized the diseases associated with LsoA and LsoB in tobacco. Then, we evaluated plant responses to these pathogens by measuring the relative expression of tobacco pathogenesis-related protein 1 gene (PR1a), a marker of salicylic acid signaling, as well as the accumulation of different metabolites, including SA, jasmonic acid, and other oxylipins at the early stages of the infection.

## 2. Materials and Methods

### 2.1. Insects

Psyllid colonies harboring LsoA and LsoB were reared on tomato plants in insect-proof cages (61 × 35 × 35 cm, Bioquip Inc., Compton, CA, USA) at room temperature with a photoperiod of 16 h of light and 8 h of darkness [24]. These colonies are henceforth referred to as LsoA and LsoB colonies, respectively. The psyllids (*Bactericera cockerelli* Sulč) used in this study were from the Western haplotype. Four potato psyllid haplotypes have been described based on a genetic marker in the mitochondrial gene COI [25,26].

### 2.2. Plant Source

All experiments were performed using tobacco plants (*N. tabacum* cv. Petite Havana SR1). Seeds were planted in Sun Gro Sunshine LP5 soil mix (Bellevue, WA, USA), and after three weeks of growth, the tobacco seedlings were transplanted individually in pots (10.5 × 10.5 cm) using the same soil mixture. The plants were fertilized once a week with the label rate of Miracle-Gro Water Soluble Tomato Plant Food (18-18-21 NPK) (Scotts Company, OH, USA) and left to grow for another two weeks. The plants were maintained on wire shelves at room temperature and a photoperiod of 16 h of light and 8 h of darkness.

### 2.3. Plant Infection

Plants were infected two weeks after transplanting when they had four fully expanded leaves. Five-week-old tobacco plants were infested with three adult psyllids from the LsoA or LsoB colonies or left free of psyllids (control). The insects were confined using an organza bag (10 × 15 cm, amazon.com) to one bottom leaf. After eight days, the leaf with the organza bag was removed from the plant to remove the insects using razor blades. The plants were maintained at room temperature on a light shelf with a photoperiod of 16 h:8 h (day:night) until the end of the experiment. The plants were kept for the duration of the experiment in rectangular 14″ × 14″ × 24″ insect cages (BioQuip, Rancho Dominguez, CA, USA). The plants were watered on average twice a week by adding water to the pan underneath the pots. This experiment was performed three times and there were at least four plants in each treatment per replicate.

### 2.4. Symptom Progression and Lso Quantification in Tobacco Plants

After infestation and psyllid removal, plants were observed weekly to record symptoms associated with Lso infection. Further, pictures were taken and samples from a top-tier leaf were collected at weeks 3, 5, and 7 postremoval of the insects.

DNA purification from the leaf samples and Lso detection were performed using the protocol previously described by Levy, et al. [27]. The presence of Lso in the plants was determined by PCR using the specific LsoF/O12 primers and the PCR conditions as described in [28]. The SSR-1 F/R primers were used to verify the Lso haplotype in the samples, as described in [9].

The copy number of Lso in an upper leaf of the plant was estimated using the standard curve method by real-time PCR [27] using the LsoF and HLBR primer set [28]. The qPCR was performed in a QuantStudio Pro System (Applied Biosystems, Thermo Fisher Scientific Inc., Waltham, MA, USA) using 50 ng of DNA and the Power UP SYBR Green Master mix (Thermo Fisher Scientific) in 96-well-plates for qPCR (VWR). The reaction volume was 10 μL. The amplification protocol was 50 °C for 2 min, 95 °C and 40 cycles of 95 °C for 15 s, and 60 °C for 30 s. The standard curve was prepared using 10-fold serial dilutions ($10^1$ to $10^6$) of the Lso 16S rDNA sequence as described in [27]. The standard dilutions of the target Lso 16S rDNA were prepared using DNA from a healthy (control) tobacco plant at a concentration of 50 ng/μL. The Lso copy number (log10 of copy number) in each sample was estimated by comparing the ΔCt values of each sample to the standard curve. Seventeen LsoA-infected plants and twenty LsoB-infected plants were analyzed.

### 2.5. Gene Expression Analysis

RNA extraction and cDNA synthesis were performed on infected and non-infected plants following the previously published method [29]. Leaf samples were collected from top-tier leaves. We evaluated the expression of the tobacco gene pathogenesis-related protein 1 (PR1a, GenBank: D90196.1) in plants infected with LsoA, LsoB, or not infected using the primers (F: 5′-TGGATGCCCATAACACAGC-3′ and R: 5′-AATCGCCACTTCCCTCAG-3′). The gene PR1a was chosen because it is regulated by salicylic acid (SA) and induced in response to many pathogens; it is a marker for systemic acquired resistance. For this analysis, the tobacco gene β-Actin (F: 5′-ATGCCTATGTGGGTGACGAAG-3′ and R: 5′-TCTGTTGGCCTTAGGGTTGAG-3′) was used as the housekeeping gene control. The gene expression analysis was performed with plant samples collected 3 and 5 weeks after removing the insects. The analysis was performed for each week using the ΔΔCt method with the uninfected sample as a reference.

### 2.6. Quantification of SA and Other Metabolites

Two independent experiments were conducted for this analysis. A total of 100 mg of leaf tissue from non-infected, LsoA-infected, and LsoB-infected tobacco plants were collected 3 weeks after insect removal. There were at least three biological replicates and six technical replicates per experiment. These samples were stored at −80 °C until the time of hormone extraction.

Hormones were extracted from leaf tissue and quantified via liquid chromatography-tandem mass spectrometry (LC-MS/MS). Tissue was homogenized with 1.0 mm dia. Zirconia beads (BioSpec Products, Bartlesville, OK, USA) and 500 μL of phytohormone extraction buffer (1-propanol/water/concentrated HCl [2:1:0.002 vol/vol/vol]) containing 500 nM internal standards of d6-SA (Sigma-Aldrich, St. Louis, MO, USA). The homogenization of samples was performed with a Precellys 24 homogenizer at a cycle of $2012 \times g \times 3$ cycles of 30 s and 30 s of rest. The samples were then agitated in the dark for 30 min at 4 °C. A total of 500 μL of dichloromethane was added to each sample and then agitated in the dark again for 30 min at 4 °C. Then, the samples were centrifuged at $21,130 \times g$ for 10 min, and the lower organic layer of each sample was transferred to a glass vial using a pipette tip and evaporated using nitrogen gas. The samples were resuspended in 150 μL of LC-MS-grade methanol and transferred to a separate tube. The samples were then stored at −20 °C for

48 h and then centrifuged again at 21,130× *g* for 4 min to pellet any debris. Approximately 100 µL of supernatant of each sample was transferred into autosampler vials for LC-MS/MS analysis. Methods were as described in [30].

LC-MS/MS analysis used an Ascentis Express C-18 column (3 cm × 2.1 mm, 2.7 µm) (Sigma-Aldrich, St. Louis, MO, USA) connected to an API 3200 LC-MS/MS (Sciex, Framingham, MA, USA) using electrospray ionization with multiple reaction mentoring. The injection volume of each sample was 10 µL, and the injection utilized a 0.5 mL/min flow rate of mobile phases A (0.2% acetic acid in water) and B (0.2% acetic acid in acetonitrile). The gradient was as follows: 0.5 min (10% B), 1 min (20% B), 21 min (70% B), 24.6 min (100% B), 24.8 min (10% B), and 29 min (stop). We quantified salicylic acid (SA), abscisic acid (ABA), and the oxylipins 9,10,11-trihydroxyoctadecenoic acid (9,10,11-THOM), 9-hydroxyoctadecadienoic (9-HOD), trihydroxy-12(Z),15(Z)-octadecadienoic acid (9-Thod), 9,10-dihydro-jasmonic acid (Dh-JA), jasmonic acid (JA), jasmonic acid isoluecine (JA-Ile), 12-hydroxy-jasmonic acid (12OH-JA), 12-Oxo-10(Z),15(Z)-phytodienoic acid (12-OPDA), azelaic acid (AZA), 13(S)-hydroxy-9(Z),11(E)-octadecatrienoic acid (13 Hod), dihydroxy-9(Z)-octadecenoic acid (DiHom), expoxy-9(Z),15(Z)-octadecenoic acid (EPOD), and 12-oxo-10(E)-octadecenoic acid (12-Koma). Other metabolites quantified were 2-hydroxy-palmitic acid (2-OH-PA) and coumaric acid (COUMA). All reference compounds used for the calibration were from Sigma-Aldrich.

### 2.7. Data Analysis

The data were analyzed using R (https://www.r-project.org/) version 4.1.2 (accessed on 6 July 2021) [31]. The Lso copy numbers in tobacco plants were log10 transformed and analyzed by the Kruskal-Wallis rank sum test, followed by pairwise comparisons using the Wilcoxon rank sum exact test with *p*-values adjusted with the Benjamini-Hochberg method [32].

Similar tests were performed to analyze PR1a expression in uninfected and LsoA- and LsoB-infected tobacco plants at weeks three and five following infections and for the metabolite accumulation at three weeks post-.

## 3. Results

### 3.1. Symptoms in Tobacco Associated with 'Candidatus Liberibacter Solanacearum' Haplotypes A and B Infection

Symptoms associated with LsoA- and LsoB-infection in tobacco plants were evaluated and compared. The first symptoms appeared on LsoB-infected plants 5 weeks after infection (after removing the insect following an inoculation access period of 1 week). The leaves of the top and middle part of the plant started to develop chlorosis and the leaves at the top of the plants were smaller and started to show curling. The plants showed a delay in growth compared to uninfected plants. At week 7, the LsoB-infected plants were stunted and the leaves in the lower tier of the plant started to die (Figure 1). Between weeks 8 and 9, all the LsoB-infected plants were dead. Symptoms associated with LsoA infection started to develop around weeks 5 and 6. The leaves showed chlorosis, reduced size on the top of the plant, and slowed growth (Figure 1). Nevertheless, LsoA-infected tobacco plants did not die.

### 3.2. Lso Quantification in Tobacco Plants

Lso was quantified in the upper leaves of the tobacco plants at different time points after the infestation (Figure 2).

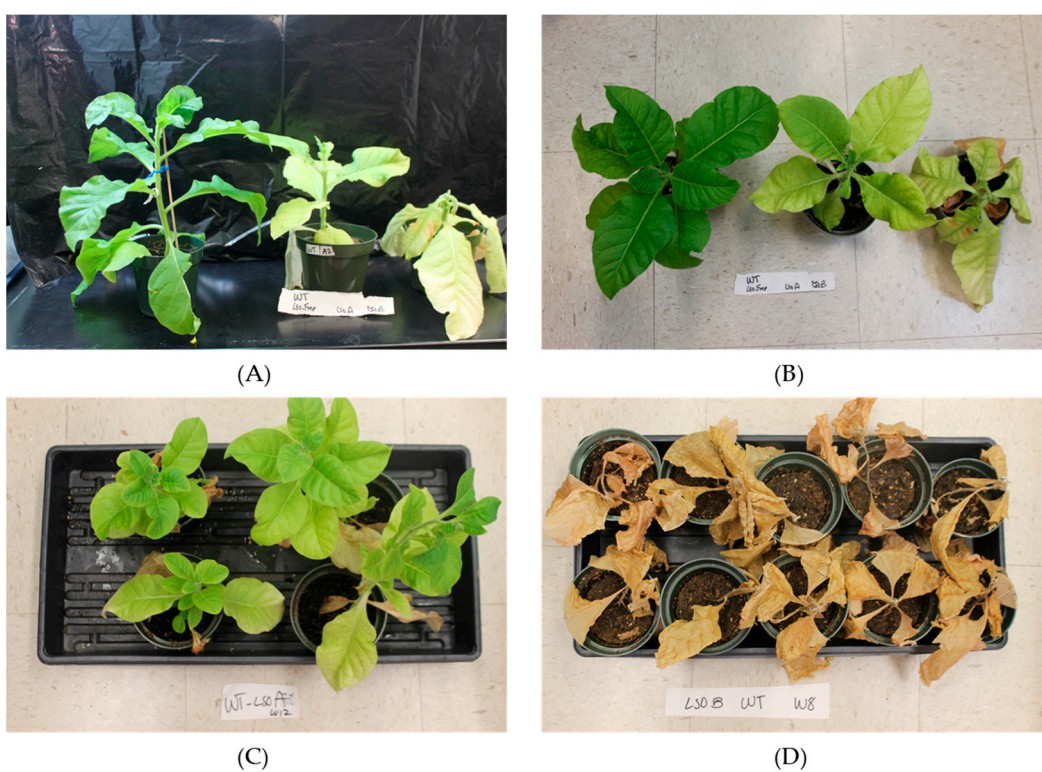

**Figure 1.** Tobacco plant symptoms. Panels (**A**,**B**): From left to right, each picture shows a non-infected, a LsoA-, and a LsoB-infected tobacco plant at week 7 post-infestation. Panel (**C**): LsoA-infected tobacco plants at week 12 post-infestation presented severe growth delay and yellowing symptoms but were alive. Panel (**D**): All LsoB-infected tobacco plants were dead.

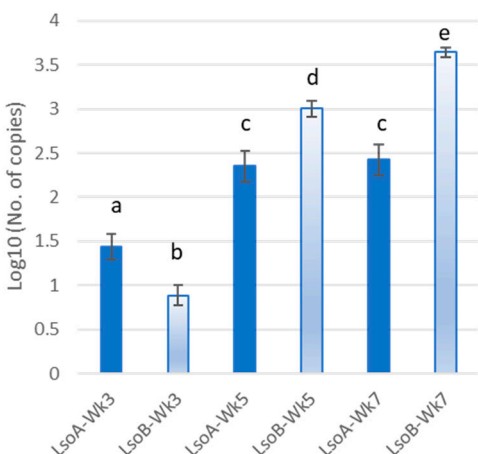

**Figure 2.** Lso quantification.

Quantification of Lso copies in tobacco leaves at 3, 5, and 7 weeks following *B. cockerelli* infestation. The values reported are log10 of Lso copy number, normalized by the standard curve method. Dark bars represent LsoA titer and lighter bars represent LsoB titer. Data represent means $\pm$ standard error of the mean. Different letters indicate statistical differences at *p*-value < 0.05 using Kruskal-Wallis rank sum test, followed by pairwise comparisons using the Wilcoxon rank sum exact test with *p*-values adjusted with the Benjamini-Hochberg method.

By week 3, both LsoA and LsoB were detectable in the tested tobacco leaves; however, differences in titer were identified (H(5) = 85.453, *p*-value < $2.2 \times 10^{-16}$). At week 3, LsoA titer was higher than LsoB, but at weeks 5 and 7, LsoA titer was significantly lower than

LsoB. While Lso B titer increased significantly up to week 7, Lso A titer plateaued at week 5, and no increase in titer was observed between week 5 and week 7.

### 3.3. PR1a Transcript Expression

The relative expression of the PR1a gene was evaluated by qRT-PCR in uninfected, LsoA-, and LsoB-infected plants at 3 and 5 weeks following infection. Infection by LsoA and LsoB induced the expression of PR1a in tobacco plants at week 3 (H(2) = 12.712, *p*-value = 0.001736) and week 5 (H(2) = 15.636, *p*-value = 0.0004025) (Figure 3). While at week 3, LsoA and LsoB induced similar expression of PR1a, this gene was upregulated by LsoB infection compared to LsoA infection in week 5.

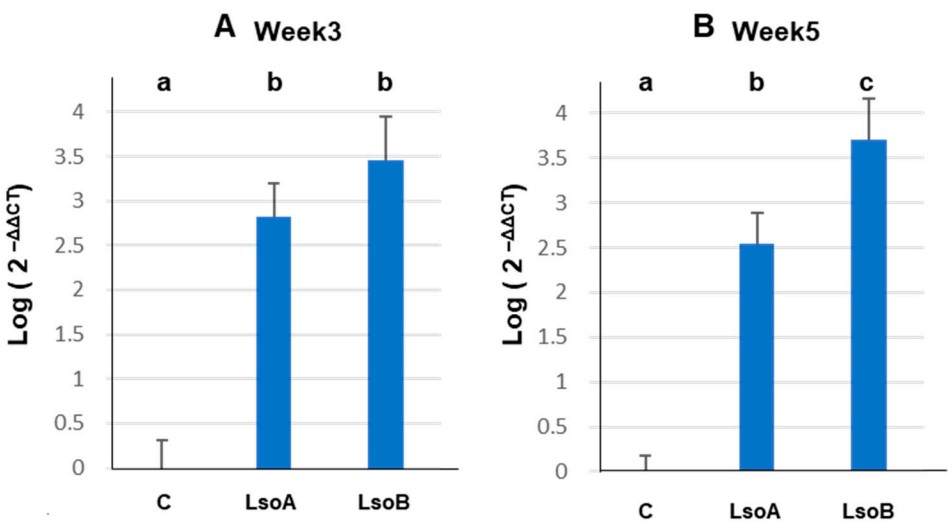

**Figure 3.** PR1a relative expression at (**A**) 3 weeks and (**B**) 5 weeks after infection.

PR1a relative expression was quantified using RT-qPCR and analyzed using the ΔΔCt method using β-Actin as the housekeeping gene control and the uninfected sample as a reference. Bars represent mean ± SEM fold change relative to uninfected plants. Different letters (a, b and c) indicate significant differences in gene expression as determined by Kruskal-Wallis analyses followed by pairwise comparisons using Wilcoxon with *p*-values adjusted with the Benjamini-Hochberg method (*p* < 0.05). The fold change in the Y-axis is in the log scale. C is for uninfected control, LsoA is for LsoA-infected, and LsoB is for LsoB-infected plants.

### 3.4. Quantification of Salicylic Acid and Other Metabolites

Hormones and secondary metabolites were quantified 3 weeks following infection. Four of the tested metabolites were found to be differentially regulated among the different treatments: SA, ABA3, and the oxylipins 9,10,11-THOM, and 9-HOD, the molecules with suspected antimicrobial activity [33].

Differences in SA concentration were measured among the treatments (H(2) = 11.061, *p*-value = 0.003964): higher SA concentration was measured in LsoB-infected plants compared to LsoA-infected or control plants (Figure 4). Similarly, differences in ABA accumulation were measured (H(2) = 8.9359, *p*-value = 0.01147), but in the case of this hormone, LsoA-infection resulted in lower accumulation compared to the control treatment; no difference was measured between LsoB-infected plants and LsoA-infected plants or control plants. Differences in 9,10,11-THOM and 9-HOD accumulation were also measured (H(2) = 7.3435, *p*-value= 0.02543 and H(2) = 11.061, *p*-value = 0.003964, respectively). 9,10,11-THOM accumulated to lower levels in LsoB-infected plants compared to controls (Figure 4), which contrasted with the increased levels of SA observed in the plants infected by this strain. The oxylipin 9-HOD accumulates at lower levels in LsoA- and LsoB-infected plants compared to the controls (Figure 4). No difference in concentration was found for the

other measured compounds: 9-Thod, Dh-JA, JA, JA-Ile, 12OH-JA, 12-OPDA, AZA, 13 Hod, DiHom, EPOD, 12-Koma, 2-OH-PA, and COUMA.

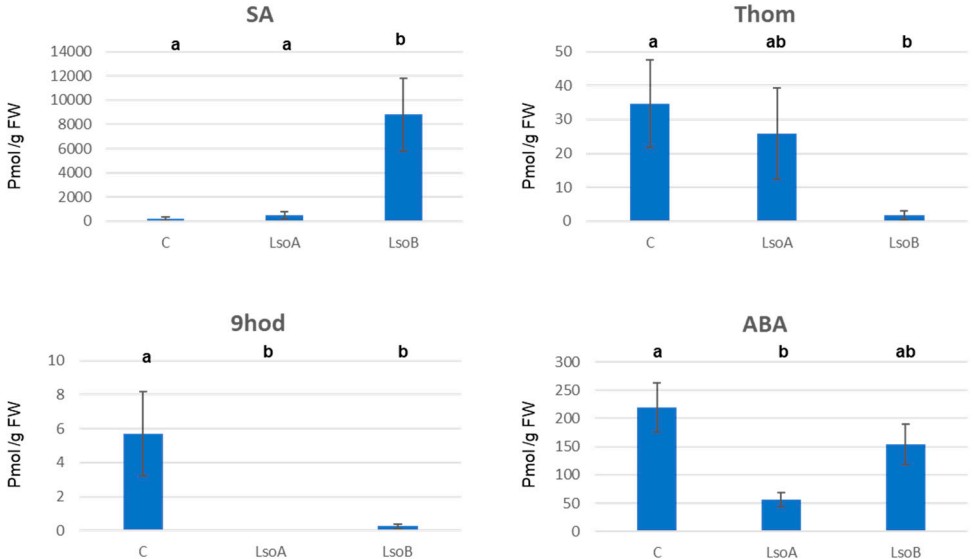

**Figure 4.** SA and other metabolites quantification.

Quantification of SA and other metabolites in uninfected (C), LsoA-, and LsoB-infected tobacco plants 3 weeks after psyllid infestation. Metabolites were measured in pmol/g of fresh weight of plant tissue. Bars represent mean ± SEM. Different letters (a and b) indicate significant differences in the accumulation of the metabolite using the Kruskal-Wallis rank sum test, followed by pairwise comparisons using the Wilcoxon rank sum exact test with *p*-values adjusted with the Benjamini-Hochberg method. The results for the following metabolites are presented: salicylic acid (SA), abscisic acid (ABA), and the oxylipins 9,10,11-trihydroxyoctadecenoic acid (9,10,11-THOM), 9-hydroxyoctadecadienoic (9-HOD).

## 4. Discussion

'*Ca.* Liberibacter' spp. is devastating bacterial plant pathogens. Progress has been made in understanding the pathogenic mechanisms and the plant responses during the infection [34–37]. However, the study of the molecular basis of the pathogen-plant interactions remains challenging because of the fastidious nature of these pathogens. The complete genome sequence of different Liberibacters has been sequenced, and genes, including putative virulence factors, have been predicted and annotated. Transcriptomic and proteomic studies examining citrus response to CLas infection have revealed that CLas actively alters multiple molecular processes in citrus [38–42]. Concomitantly, transcriptomic studies in potato and tomato have highlighted several signaling pathways involved in immunity in response to Lso, such as ethylene or jasmonic acid signaling and defense proteins [21,29,43]. The Lso-potato psyllid-solanaceous plant pathosystem offers advantages to help elucidate the mechanisms at play during plant infection with Liberibacters. Some of these advantages are the existence of Lso haplotypes infecting the same hosts but with differences in virulence [18–20,24] and the existence of genetic resources for plants susceptible to Lso, such as tobacco and tomato.

Indeed, tobacco is a model plant to study plant−pathogen interactions, and it is also a major cash crop in some countries affected by Lso. The disease caused by Lso in tobacco was described in 2013 in Nicaragua and Honduras [22,23]. In both countries, Lso and potato psyllids were found to be associated with tobacco plants showing aerial symptoms similar to those associated with zebra chip [22,23]. However, the diseases associated with each of the two main Lso haplotypes, LsoA and LsoB, have not been characterized, and it is not known whether infection by each haplotype results in similar diseases, such as

in potato [19], or if differences in the diseases caused by these haplotypes exist such as in tomato [20,21]. In the current study, we describe the symptoms associated with LsoA and LsoB in tobacco plants. Similar symptoms were associated with LsoA and LsoB infection; those included chlorosis, stunting, leaf curling, and reduced leaf size. However, plants infected with LsoB died prematurely, whereas those infected with LsoA did not. We also compared the titer of each haplotype in the top leaves of the plant over time. We have previously shown that in tomato and potato, Lso is first detectable in the upper leaves of the plant 3 weeks after psyllid infestation [27]. Here, we showed that also in tobacco, Lso could be detected in the top leaves 3 weeks after plant infestation. Three weeks is a relatively early time point during the infection; no visual symptoms had yet developed in the infected plants. Interestingly, while at week 3, LsoA titer was higher than LsoB titer in the infected plants, LsoB titer was higher than LsoA titer at weeks 5 and 7. These trends are similar to those described in tomato [20] but not potato [17,18]. In tomato and tobacco, both the development of symptoms and the bacterial titer were different between LsoA- and LsoB-infected plants. While the LsoB titer increased over time, LsoA titer plateaued after 5 weeks. Furthermore, similar to tomato, tobacco plants infected with LsoB died prematurely, whereas those infected with LsoA did not. Therefore, tobacco is a host of interest in identifying key Lso proteins involved in infection because it could be used to link the genetic differences between LsoA and LsoB and the different outcomes of the infection.

Liberibacter pathogenicity and virulence factors remain largely unknown. Effectors are key factors of pathogenicity during the interaction between plants and pathogens [44]. Prasad, et al. [45] identified close to 100 Liberibacter proteins potentially secreted by the SEC system; some of these are candidates for acting as bacterial effectors, and the plant responses to their transient expression were analyzed. Recently, Thapa, De Francesco, Trinh, Gurung, Pang, Vidalakis, Wang, Ancona, Ma, and Coaker [6] proposed that only 30 of these proteins are involved in regulating plant defense mechanisms. While a larger number of CLas effectors have been characterized and tested, fewer Lso proteins have been tested for their potential role as effectors [35,46–48].

In an effort to elucidate the differences in pathogenicity between LsoA and LsoB in association with tobacco, first, we evaluated the expression of PR1a following infection. Because plants become infected following transmission by psyllids, we compared the effect of herbivory and infection to control plants. PR1a was induced in the infected plants at 3 and 5 weeks postinfection. Here, the PR1a upregulation measured was in response to both the pathogen and the vector because the control plants were not infested with Lso-free psyllids. While similar levels of upregulation were measured in LsoA- and LsoB-infected plants at week 3, the PR1a expression was higher in plants infected with LsoB than LsoA at week 5. This change in gene expression might be linked to the difference in plant colonization between the two Lso haplotypes; while LsoB titer continued to increase, LsoA titer plateaued after week 5. The difference in pathogenicity between LsoA and LsoB, resulting in differences in bacterial titer and symptom development, is probably also linked with differences in plant responses.

Interestingly, Ibanez, et al. [49] showed that citrus PR1 was induced following 14 and 150 days of herbivory by the Asian citrus psyllid, *Diaphorina citri*, but not after 7 days of exposure to the insect feeding. In that study, SA accumulated at higher levels only in plants exposed to 150 days of herbivory but not after 7 or 14 days. Therefore, we also evaluated the accumulation of SA in the tobacco-infected samples.

Here, we found a significant accumulation of SA at week 3 in LsoB- but not in LsoA-infected plants. The accumulation of SA in Lso-infected tobacco plants was expected since PR1a was upregulated in the infected plants, and the downstream signaling and the events that occur in plants in response to SA play a central role in plant defense against pathogens in many plant species [50]. Indeed, the immune responses mediated by SA are important components of both PAMP-Triggered Immunity (PTI) and Effector-Triggered Immunity (ETI) and are also key for the triggering of systemic acquired resistance [51,52] and defense against (hemi)biotrophic pathogens [53]. Further, SA biosynthesis inhibition



or blocking its accumulation can enhance plant susceptibility to some pathogens [52]. In 2017, Li, et al. [54] characterized a monooxygenase FAD-binding protein (CLIBASIA_00255) from CLas as a salicylic acid hydroxylase. This gene is present in all sequenced pathogenic '*Ca*. Liberibacter' species and the homologs share a high degree of similarity. Li and collaborators demonstrated that CLas Sah is a functional SA hydroxylase that degrades SA and suppresses plant defenses. They also showed that the application of SA analogs slowed the progression and severity of HLB disease by neutralizing the effect of SA hydroxylase in degrading SA [54].

In the present study, both LsoA and LsoB infection induced the expression of PR1a, but only LsoB-infected plants had accumulated higher levels of SA 3 weeks after infection. Although PR1a expression and SA are directly linked, in these experiments, SA was not found to be significantly induced in LsoA-infected plants.

It is tempting to link the difference in SA accumulation at week 3 with the difference in PR1a induction and the differences in Lso titer increase measured at week 5. Based on the results presented here, it appears that in tobacco, LsoA is less pathogenic than LsoB and the infection with this bacterial haplotype results in the development of milder symptoms. This difference in virulence results in the induction of different plant responses, including the differential accumulation of SA as early as 3 weeks after infection.

Another difference identified in this experiment was the lower accumulation of ABA in the LsoA-infected plants compared to the control plants. This reduced ABA accumulation probably resulted in the closure of the stomata and reduced water loss in LsoA-infected plants [55]. While the accumulation of ABA between LsoA- and LsoB-infected plants was not significantly different, LsoB-infected plants appeared wilted compared to the LsoA-infected plants: this is observable at 7 weeks postinfection in Figure 1.

The oxylipin 9-HOD accumulated at higher levels in control plants than in LsoA- and LsoB-infected plants. This lower accumulation in infected plants is not consistent with the expected positive contribution of this oxylipin to defenses. While in some studies JA signaling was identified as involved in plant response to Lso and psyllids [29], in this study, no difference between control and infected plants was observed in the accumulation of JA or its biologically active form, JA-Ile, or the JA precursor 12-OPDA. The implication of these results for plant infection deserves further exploration. These analyses were performed 3 weeks after infection to evaluate differences after infection but before symptoms developed. We expect that other metabolites could accumulate differentially at other time points following infection.

**Author Contributions:** Conceptualization, J.G.L. and C.T.; methodology, J.G.L., A.M.-H, N.M. and K.M.B.-F.; formal analysis J.G.L., N.M., M.V.K. and C.T.; investigation, A.M.-H. and K.M.B.-F.; writing, J.G.L. and C.T.; supervision C.T.; funding acquisition, J.G.L., M.V.K. and C.T. All authors have read and agreed to the published version of the manuscript.

**Funding:** This research was funded by Texas A&M University and Texas A&M AgriLife Research (Controlling Exotic and Invasive Insect-Transmitted Pathogens) and USDA National Institute of Food and Agriculture, Hatch/Multi-State project 1013013 and Hatch project TEX0-9934 Accession Number 7002235. Research in Kolomiets laboratory is supported by 2021-67013-33568 grants awarded to M.V.K.

**Institutional Review Board Statement:** Not applicable.

**Informed Consent Statement:** Not applicable.

**Data Availability Statement:** The data presented in this study are available in this article.

**Conflicts of Interest:** The authors declare no conflict of interest.

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
