# Peer review of "Evaluation of the Effect of ‘Candidatus Liberibacter Solanacearum’ Haplotypes in Tobacco Infection"

_agronomy, doi:10.3390/agronomy13020569_

Round 1

Reviewer 1 Report

‘Candidatus Liberibacter solanacearum’ (Lso) is a bacterial plant pathogen widely infecting the solanaceous plant. In the manuscript “Evaluation of the effect of a ‘Candidatus Liberibacter’ haplotypes in tobacco infection” detect two predominant haplotypes of Lso, LsoA and LsoB’ virulence on their host Tobacco, Nicotiana tabacum. Also they found SA was regulated by LsoB but not LsoA whereas ABA was regulated by LsoA but not LsoB. I just have one suggestion. The error bar is too long in figure 3 and figure 4. Could the author please include more measure data?

Author Response

The error bar is too long in figure 3 and figure 4. Could the author please include more measure data?

 We corrected the figure legend and the graph. The previous graphs were box plots therefore the bars were not error bars, but we mislabeled them in the legend. We are presenting bar charts now like that there is less confusion.

Reviewer 2 Report

The authors evaluated the infection of tobacco with LsoA and LsaB. Are SA and other oxylipins ABA synthesis pathway genes regulated? Add this related experiment. Draw a clear Figures in the revised manuscript and add detailed Figures Legend.

Author Response

Are SA and other oxylipins ABA synthesis pathway genes regulated? Add this related experiment.  The main objective was to evaluate the symptoms associated with each haplotype. The metabolite work is exploratory. Now we will focus on that but it is not part of this manuscript.   

Draw a clear Figures in the revised manuscript We did new figures 3 and 4

and add detailed Figures Legend.  We edited the figure legends

Reviewer 3 Report

In this study, Levy et al characterized the infection associated with two haplotypes of  ‘Candidatus Liberibacter solanacearum’ (Lso) in tobacco. They found that LsoB caused more severe symptoms than LsoA, and LsoA and LsoB titers were dynamic during the 7 weeks of experiment under laboratory conditions. They also measured SA and other oxylipins at early time point of infection and found SA was regulated by LsoB but not LsoA whereas ABA was regulated by LsoA but not LsoB. This is a well organized manuscript, I recommend to accept it after following revisions:

1.State further the relationship between Lso and CLas in line 31-33.

2. Bactericera cockerelli in line 41 should be italic.

3. It is enough using three adult psyllids for five-week-old tobacco plants in line 70-71.

4. Adding a reference for amplification protocol in line 88-89.

5. Also adding a reference for Hormones extraction in line 112-113.

6. What time “All LsoB-infected tobacco plants were dead” in line 170.

7. For figure 4, why did you only detect the SA and other metabolites  3 weeks after psyllid infestation?

Author Response

Reviewer 3

In this study, Levy et al characterized the infection associated with two haplotypes of  ‘Candidatus Liberibacter solanacearum’ (Lso) in tobacco. They found that LsoB caused more severe symptoms than LsoA, and LsoA and LsoB titers were dynamic during the 7 weeks of experiment under laboratory conditions. They also measured SA and other oxylipins at early time point of infection and found SA was regulated by LsoB but not LsoA whereas ABA was regulated by LsoA but not LsoB. This is a well organized manuscript, I recommend to accept it after following revisions:

1.State further the relationship between Lso and CLas in line 31-33. Both are plant pathogens vectored by psyllids. Although they have evolved reduce genomes as obligatory symbionts, the common ancestor of CLas and Lso is estimated to be around 100mya (https://bsppjournals.onlinelibrary.wiley.com/doi/pdfdirect/10.1111/mpp.12925). We modified the article line 35

  1. Bactericera cockerelli in line 41 should be italic. Corrected

  1. It is enough using three adult psyllids for five-week-old tobacco plants in line 70-71. Yes this is a method we have used consistently to infect other solanaceous plants (published in several manuscripts) and here we showed that it is also enough to infect tobacco plants.

  1. Adding a reference for amplification protocol in line 88-89. Reference 24 is the reference for PCR

  1. Also adding a reference for Hormones extraction in line 112-113. We added reference 29

  1. What time “All LsoB-infected tobacco plants were dead” in line 170. Now line 192 Between weeks 8 and 9, all the LsoB-infected plants were dead.

  1. For figure 4, why did you only detect the SA and other metabolites 3 weeks after psyllid infestation? We decided to conduct the test at the early time point, before symptoms develop. We know that week 3 is the earlier we are able to detect Lso in the aerial parts of potato and tomato plants. For this analysis we wanted the infection to be underway to ensure that we would see differences, but we wanted to perform the studies before the plants developed symptoms. Therefore, in our studies we decided that was the time point for early onset of the disease. Week 6-7 the disease might be too advance to detect any change that might be meaningful Week 8-9 Lso B plants are dead. This justification was added to the discussion (line 374).

Reviewer 4 Report

The presented study is well done and presents the suitability of tobacco as a model plant for studying virulence and plant pathogen interactions with Candidatus Liberibacter solanacearum. The overall quality is fine but there are plenty of typos and some citation formats to correct. Furthermore, as a reader, I would appreciate if more methodological details will be given (see my comments below).

Here below some mistakes found. Another proofeading is strongly recommended.

Line 24: prepositions missing «but not by LsoA», “were reduced in LsoA- but not in LsoB-infected”

Line 65: please explain and justify why you studied early regulation of oxylipins.

Lines 73-77: Please indicate species and origin of psyllids. Explain Western haplotype here or in the introduction

Line 86: at room temperature

Line 92: please indicate supplier and size of organza bags

Line 102 : how the psyllid were removed? Mechanically or with an insecticidal spray?

Line 106/116: Levy et al. (2011) is the wrong citation format and the reference is missing in the reference list

Line 109: Lin et al. (2011) is the wrong citation format and the reference is missing in the reference list

Line 113: please indicate supplier of Power UP SYBR Green Master mix

Line 114: please indicate reaction volume and plate type used

Line 118: please indicate amplification efficiency, regression of the standard curve and the LOD of the assays. What source of Lso gene copy number control did you use? Plasmid DNA? Please give more details.

Line 131: beta-Actin was used as house-keeping gene control

Line 134: as a reference

Line 145 what was the concentration of the HCl?

Line: 151: how did you collect the lower organic layer? With a long pipette tip or a needle?

Line: 160 please add another subject (flow rate) into the sentence. The injection volume can not utilize a 0.5mL/min flow rate

Lines 163-171: please indicate supplier of the reference compounds used for the calibration.

Line 174: please indicate R version and reference

Line 177: please indicate references for the statistical models used.

Line 214: B cockerelli in italics

Line 250-251: Fig. 4, in all 4 graphs: please center the small letters indicating significant differences

Line 303/305/322: Citation format not correct Prasad (2016), Thapa 2020, Ibanez 2019

Author Response

Reviewer 4

The presented study is well done and presents the suitability of tobacco as a model plant for studying virulence and plant pathogen interactions with Candidatus Liberibacter solanacearum. The overall quality is fine but there are plenty of typos and some citation formats to correct. Furthermore, as a reader, I would appreciate if more methodological details will be given (see my comments below).

Here below some mistakes found. Another proofeading is strongly recommended.

Line 24: prepositions missing «but not by LsoA», “were reduced in LsoA- but not in LsoB-infected” this was Changed thanks

Line 65: please explain and justify why you studied early regulation of oxylipins. We decided to conduct the test at the early time point, before symptoms develop. We know that week 3 is the earlier we are able to detect Lso in the aerial parts of potato  and tomato plants. For this analysis we wanted the infection to be underway to ensure that we would see differences, but we wanted to perform the studies before the plants developed symptoms. Therefore, in our studies we decided that was the time point for early onset of the disease. Week 6-7 the disease might be too advance to detect any change that might be meaningful Week 8-9 Lso B plants are dead. This justification was added to the discussion (line 374).

Lines 73-77: Please indicate species and origin of psyllids. Explain Western haplotype here or in the introduction  

Response The psyllids (Bactericera cockerelli Sulč) used in this study are from the Western haplotype. Four psyllids’ haplotypes have been described based on a genetic marker in the mito-chondrial gene COI.

Line 86: at room temperature Corrected thanks

Line 92: please indicate supplier and size of organza bags (10 x 15 cm, amazon.com)

Line 102 : how the psyllid were removed? Mechanically or with an insecticidal spray? The infested leave was cut after a week. This is in the plant infection section line 93

Line 106/116: Levy et al. (2011) is the wrong citation format and the reference is missing in the reference list this is corrected thanks

Line 109: Lin et al. (2011) is the wrong citation format and the reference is missing in the reference list this is corrected thanks

Line 113: please indicate supplier of Power UP SYBR Green Master mix (Thermo Fisher Scientific Inc.)

Line 114: please indicate reaction volume The reaction volume was 10 µL. and plate type used (96 well plate for qPCR VWR)

Line 118: please indicate amplification efficiency, regression of the standard curve and the LOD of the assays. What source of Lso gene copy number control did you use? Plasmid DNA? Please give more details. All this information is described in details in the cited reference levy et al 2011 reference 24  

Line 131: beta-Actin was used as house-keeping gene control corrected thanks

Line 134: as a reference corrected thanks

Line 145 what was the concentration of the HCl? We added concentrated in the text. Because we used the commercial HCl.  

Line: 151: how did you collect the lower organic layer? With a long pipette tip or a needle? We added using a pipette tip

Line: 160 please add another subject (flow rate) into the sentence. The sentence was changed for clarification

Lines 163-171: please indicate supplier of the reference compounds used for the calibration. All reference compounds used for the calibration were from Sigma-Aldrich.

Line 174: please indicate R version and reference (https://www.r-project.org/) version R version 4.1.2 (2021-11-01) [27]

Line 177: please indicate references for the statistical models used.  We added reference

Line 214: B cockerelli in italics corrected

Line 250-251: Fig. 4, in all 4 graphs: please center the small letters indicating significant differences corrected

Line 303/305/322: Citation format not correct Prasad (2016), Thapa 2020, Ibanez 2019 corrected